# Noninvasive Models to Assess Liver Inflammation and Fibrosis in Chronic HBV Infected Patients with Normal or Mildly Elevated Alanine Transaminase Levels: Which One Is Most Suitable?

**DOI:** 10.3390/diagnostics14050456

**Published:** 2024-02-20

**Authors:** Shasha Ma, Lian Zhou, Shutao Lin, Mingna Li, Jing Luo, Lubiao Chen

**Affiliations:** Department of Infectious Diseases, The Third Affiliated Hospital of Sun Yat-Sen University, Guangzhou 510630, China; mashsh8@mail2.sysu.edu.cn (S.M.); zsdxzl@163.com (L.Z.); linsht25@mail2.sysu.edu.cn (S.L.);

**Keywords:** CHB, noninvasive model, liver fibrosis, inflammation, APGA

## Abstract

The prevalence of substantial inflammation or fibrosis in treatment-naïve patients with chronic hepatitis B (CHB) and normal alanine transaminase (ALT) levels is high. A retrospective analysis was conducted on 559 consecutive patients with hepatitis B virus infection, who underwent liver biopsy, to investigate the value of noninvasive models based on routine serum markers for evaluating liver histology in CHB patients with normal or mildly elevated ALT levels and to provide treatment guidance. After comparing 55 models, we identified the top three models that exhibited excellent performance. The APGA model, based on the area under the receiver operating characteristic curve (AUROC), demonstrated a superior ability to evaluate significant (AUROC = 0.750) and advanced fibrosis (AUROC = 0.832) and demonstrated a good performance in assessing liver inflammation (AUROCs = 0.779 and 0.874 for stages G ≥ 2 and G ≥ 3, respectively). APGA also exhibited significant correlations with liver inflammation and fibrosis stage (correlation coefficients, 0.452 and 0.405, respectively (*p* < 0.001)). When the patients were stratified into groups based on HBeAg status and ALT level, APGA consistently outperformed the other 54 models. The other top two models, GAPI and XIE, also outperformed models based on other chronic hepatitis diseases. APGA may be the most suitable option for detecting liver fibrosis and inflammation in Chinese patients with CHB.

## 1. Introduction

The hepatitis B virus (HBV) infection continues to pose a global public health challenge, despite the decreasing trend in HBV infection rates over the years due to widespread access to the hepatitis B vaccination and preventive measures. However, according to the World Health Organization (WHO), an estimated 296 million individuals were living with chronic hepatitis B (CHB) in 2019, with approximately 1.5 million new infections occurring annually. Moreover, there were an estimated 820,000 deaths primarily caused by cirrhosis and hepatocellular carcinoma (HCC), with a significant proportion of these fatalities occurring in the Asia–Pacific region [1].

Previous studies suggested that patients in the immune-tolerant (IT) phase have slow disease progression due to little inflammation or fibrosis in the liver [2,3]. Numerous studies have shown that one-third to one-half of treatment-naïve CHB patients with normal alanine transaminase (ALT) levels may still experience substantial inflammation or fibrosis and may even have increased risks of HCC and death/transplantation [4]. Consequently, these patients require antiviral therapy, which has exhibited comparable efficacy [5,6]. Based on these findings, the indications for antiviral therapy have recently been expanded [7,8], aiming to prevent unnecessary deaths through early intervention in selected IT-phase patients [4]. Nevertheless, IT-phase patients continue to exhibit poor rates of seroconversion after receiving antiviral therapy and are more prone to developing treatment resistance [9]. Therefore, determining the state of liver histology at the commencement of treatment is crucial [3].

Although histological examination has long been regarded as the gold standard for assessing liver inflammation and fibrosis, the risks and costs associated with liver biopsy have limited its widespread use. Individuals with normal ALT levels are often hesitant to undergo this invasive procedure. Consequently, noninvasive methods for predicting liver fibrosis, including noninvasive models, transient elastography (TE), two-dimensional shear wave elastography (2D-SWE), and the FibroTest, have been proposed over the past few decades. Noninvasive and repeatable assessments of liver fibrosis are widely facilitated by TE, FibroTest, and 2D-SWE [10,11,12]. However, the diagnostic accuracy of these tests may be influenced by the operator’s expertise, the absence of extensively validated cutoff values for specific stages of fibrosis, increasing rates of unreliability in patients with higher obesity, and the high cost of the equipment. These factors could limit the clinical utility of these tests in primary hospitals [13,14]. Given these findings, numerous noninvasive models based on serum markers, including the widely recommended Fibrosis-4 (FIB-4) and aminotransferase-to-platelet ratio index (APRI), have been developed and extensively discussed in the past two decades [15,16,17,18,19]. Furthermore, a majority of the serum-based noninvasive models were developed primarily using patients with chronic hepatitis C (CHC), while only a few were established using CHB cohorts. Additionally, other than FIB-4 and APRI, which are endorsed by guidelines or general consensus, no models have demonstrated satisfactory outcomes in clinical practice. Consequently, an increasing number of novel noninvasive models and nomograms, such as the IT [20], PAPAS [21], APRG [22], APGA [23], and Chen models, have also emerged for assessing fibrosis in CHB patients [24].

In this study, our objective was to evaluate the diagnostic accuracy of non-patented, noninvasive models, nomograms or indices based solely on routine serum biomarker data. These models are suitable for implementation in almost any medical facility and can effectively discriminate liver fibrosis and cirrhosis in both treatment-naïve individuals (including those with ALT < 2× the upper limit of normal [ULN]) and treated CHB patients who have discontinued antiviral therapy in China. The findings from this study may assist healthcare professionals in making appropriate decisions regarding antiviral treatment and potentially reduce the necessity for liver biopsy among CHB patients with normal or mildly elevated ALT levels through the utilization of a highly suitable model.

## 2. Materials and Methods

### 2.1. Patients

From January 2017 to December 2022, a total of 844 patients with CHB who had undergone liver biopsy at the Third Affiliated Hospital of Sun Yat-sen University were included in this study. The criteria for eligibility were as follows: (1) chronic HBV infection for over 6 months; (2) ALT and/or aspartate transaminase (AST) levels less than 2 times the ULN (ALT and AST ULN = 40 U/L) [25] at least 6 months before enrolling; (3) patients who had not previously received antiviral therapy, had received a transient course of antiviral therapy, or had stopped antiviral therapy for more than a year while HBV-DNA remained positive; (4) age of 16–70 years old; and (5) the liver biopsy tissue met the immunohistochemical requirements with an available pathological report. The overall exclusion of 245 patients was based on the following criteria: (1) ALT and/or AST levels exceeding 2 times the ULN; (2) did not discontinue antiviral therapy prior to liver puncture; (3) liver disease of other etiologies, such as viral coinfection, autoimmune hepatitis (AIH), primary biliary cholangitis (PBC), primary sclerosing cholangitis (PSC), alcohol-associated liver disease (ALD), or nonalcoholic fatty liver disease (NAFLD); (4) liver cirrhosis (LC) or carcinoma; (5) aged > 70 years or younger than 16 years; (6) systemic diseases affecting the liver, such as HIV infection, heart failure, or hyperthyroidism; and (7) insufficient data availability.

A total of 599 patients who met the eligibility criteria were included in the final study cohort. Among them, 514 patients did not receive any antiviral treatment, while 85 patients had received nucleoside analogs (NAs) and/or peg-IFN antiviral therapy but had discontinued it for more than 1 year before liver puncture. Additionally, out of these patients, 221 were HBeAg positive, 377 were HBeAg negative, and one patient had missing HBeAg data. The detailed study design is illustrated in Figure 1.

The clinical data of eligible patients were retrospectively collected either on the day of liver biopsy or one week prior. This comprehensive dataset included demographic information, antiviral therapy history, and routine serum marker data commonly used in medical facilities. These markers encompassed the liver function indicators (such as ALB, GLO, TBIL, DBIL, ALT, AST, GGT, and ALP); coagulation function markers (e.g., PT, PTA, and INR); total cholesterol levels; apolipoprotein A1 levels; glucose levels; complete blood count values; and serological markers for HBV infection, including HBsAg, HBeAg, anti-HBe, anti-HBc, HBV DNA load and AFP levels. All of the above biomarkers were assessed at the Clinical Laboratory of the Third Affiliated Hospital of Sun Yat-sen University. Additionally, imaging examinations such as ultrasound scans or CT/MR scans were performed in the Radiology and Ultrasound Department to exclude cirrhosis and hepatocellular carcinoma (HCC).

### 2.2. Liver Histological Examination

Liver biopsies were performed via the percutaneous echo-assisted technique, with a minimum requirement of 6 portal tracts. The slides were examined and interpreted by two pathologists at the Third Affiliated Hospital of Sun Yat-sen University. Biopsies were categorized into stages based on the Scheuer scoring system [26]: G 0–4 and S 0–4. In this study, significant fibrosis (SF) and advanced fibrosis (AF) were defined as pathological stages ≥ S2 and ≥S3, respectively. Significant inflammation and severe inflammation were defined as pathological stages ≥ G2 and ≥G3, respectively.

### 2.3. Noninvasive Models

A literature search was conducted in the PubMed, Chinese National Knowledge Infrastructure (CNKI), and Web of Science databases and the relevant literature citations from January 2000 to December 2022 were noted to identify non-patented liver fibrosis models, nomograms, and indices utilizing serum markers. These models have been previously used for assessing fibrosis in patients with viral hepatitis B or C, PBC, or NAFLD, among others. Initially, more than ninety noninvasive models were collected; however, only 55 were included after excluding models that lacked important indices, such as haptoglobin, hyaluronic acid (HA), matrix metalloproteinase-1 (MMP-1), procollagen III N-terminal peptide (PIIINP), soluble CD163 (sCD163), and α2-macroglobulin (α2-MG); the quantification of HBeAg and anti-HBc antibodies; and the liver stiffness measure (LSM) by transient elastography [24,27,28,29,30,31]. The included models are summarized in Appendix A.

### 2.4. Statistical Analysis

The data were analyzed using SPSS software version 20.0 (IBM Corp, Armonk, NY, USA) and GraphPad Prism Software version 8.0 (GraphPad Software). The one-sample Kolmogorov–Smirnov test was used for a normality analysis, and the data are presented as mean values ± standard deviations or median values with interquartile ranges (P25, P75). Student’s *t* test and the Mann–Whitney U test were used for group comparisons. Categorical variables were expressed as numbers and percentages and were compared using the chi-square test. Areas under the receiver operating characteristic curve (AUROCs) with 95% confidence intervals (CIs) were utilized to evaluate the performance of the noninvasive models in diagnosing liver inflammation and fibrosis. Correlation analysis between histopathological fibrosis stage and liver inflammation was conducted using either Pearson’s or Spearman rank test based on the normality of the distribution of the parameter in question. Optimal cutoffs were determined by maximizing Youden’s index. The diagnostic metrics included sensitivity, specificity, positive predictive value (PPV), negative predictive value (NPV), positive likelihood ratio (PLR), and negative likelihood ratio (NLR). A *p* value less than 0.05 (2-tailed) indicated statistical significance.

To comprehensively compare the accuracy of the noninvasive models, we adopted a grading system derived from Dong’s study [32] (Table 1) and used it to assess both fibrosis and inflammation. A model with an AUROC below 0.700 scored 0, 0.700~0.750 scored 1, 0.750~0.800 scored 2, and >0.800 scored 3. The final score for each model was determined by summing the scores for S ≥ 2, S ≥ 3, G ≥ 2, and G ≥ 3. Subsequently, a model scoring 0–4 was designated Grade C, representing low diagnostic efficiency; a model scoring 5–8 was designated Grade B, representing moderate diagnostic efficiency; and a model scoring 9–12 was designated Grade A, representing high diagnostic efficiency. The grading system was also applied to validate liver inflammation and liver fibrosis based on the HBeAg status or ALT levels.

## 3. Results

### 3.1. Patient Characteristics

The study design is illustrated in Figure 1. From January 2017 to December 2022, a total of 844 CHB patients who had undergone liver biopsy were recruited for this study. After applying the eligibility criteria, a cohort of 599 patients were enrolled as study subjects for subsequent analysis. There were 514 patients who were not treated with any NAs or peg-IFN, while 85 patients were treated with NAs and/or peg-IFN but had discontinued them for more than 1 year before liver puncture. According to the Scheuer scoring system, S ≥ 2 and/or G ≥ 2 were defined as significant pathological changes in liver injury (SHCHI), and we divided patients into two groups. In our cohort, 221 (35.2%) patients were HBeAg positive; 118 (31.4%) patients had a fibrosis stage of S ≥ 2, and 64 (10.7%) patients had a fibrosis stage of S ≥ 3, while 191 (34.2%) patients had an inflammation grade of G ≥ 2 and 50 (8.9%) patients had an inflammation grade of G ≥ 3, as shown in Figure 2. The log (HBsAg) value was 3.32 (2.86, 3.91) in the S0–1 group and 3.43 (3.0, 3.81) in the S2–4 group (*p* = 0.656), and the log(HBV DNA) values were 4.425 (3.2975, 7.8025) and 5.16 (3.675, 6.11), respectively (*p* = 0.773); moreover, there were no significant differences between these two groups. The comprehensive demographic and laboratory parameters of the subjects are presented in Table 2.

### 3.2. Validation of Noninvasive Models in All Patients

Due to the limited sample size in the S4 group, potential bias may exist. Therefore, a comparison of models was conducted between patients with and without SF (S ≥ 2) and AF (S ≥ 3) across the entire population. The AUROCs for each model for discriminating between SF and AF were calculated and are summarized in Table 3.

In general, the 55 noninvasive models were found to be effective at diagnosing both SF and AF among all patients, regardless of their HBeAg status or AST/ALT levels. However, not all the models demonstrated satisfactory performance. In the discrimination of SF among all patients, four models exhibited AUROCs higher than 0.700: the APGA (0.750), APPCI (0.726), GAPI (0.719), and Xie models (0.714). For identifying AF, the AUROC of the APGA model (0.832) exceeded 0.800, while the three models with the next highest AUROCs were GAPI (0.802), Xie model (0.80), and S index (0.797). Overall, each noninvasive model generally displayed a greater AUROC for diagnosing AF than for diagnosing SF, as presented in Table 3 and Figure 3.

Most noninvasive models have been utilized for discriminating liver fibrosis stages but have limited application in validating inflammation. Hence, our study aimed to validate the efficacy of these models in distinguishing liver inflammation. To compare the performance of different models between patients with and without G ≥ 2 and G ≥ 3, we calculated the AUROCs for each model; these values are summarized in Table 3 and Figure 3.

The four models with the highest AUROCs for identifying G ≥ 2 in all patients were APGA (0.779), Xie model (0.744), AGAP (0.733), and Logit(Y) (0.725); the AUROC of GAPI was 0.705. For discriminating G ≥ 3, the four models with the highest AUROCs were APGA (0.874), Xie model (0.861), Wang I (0.859), and AGAP (0.857); the AUROC of GAPI was 0.834. Generally, there was a gradual increase in the AUROC of each noninvasive model for diagnosing inflammation as the inflammation grade increased, as shown in Table 3 and Figure 3.

### 3.3. Evaluation and Comparison of Noninvasive Models in the HBeAg-Negative and HBeAg-Positive Groups

The patients were stratified into HBeAg-positive and HBeAg-negative groups for comparative analysis and further validation. In general, the noninvasive models exhibited higher AUROCs in the HBeAg-positive group than in the HBeAg-negative group for discriminating each fibrosis stage, indicating that these models may be more appropriate for managing HBeAg-positive patients.

In the HBeAg-positive group, the three noninvasive models with the highest discriminatory ability for SF were the Gao-1 model (AUROC = 0.77), the Wang I model (AUROC = 0.766), and the APGA model (AUROC = 0.761). The AUROCs of GAPI and the Xie model were 0.756 and 0.67, respectively, indicating moderate discriminatory performance for SF. For AF discrimination in this group, the top three models were GAPI (AUROC = 0.859), Gao-1 (AUROC = 0.851), and APGA (AUROC = 0.846). The Xie model had an AUROC of 0.769 for AF discrimination(Table 3).

In the HBeAg-negative group, three different models with AUROCs above 0.700 were identified for each fibrosis stage discrimination. For SF discrimination, the models with the best discrimination were APGA (AUROC = 0.745), APPCI (AUROC = 0.757), and Mehdi’s model (AUROC = 0.718). For AF discrimination, the top three models were APGA (AUROC = 0.824), the XIE model (AUROC = 0.816), and GUCI (AUROC = 0.787) (Table 3).

When evaluating the performance of these noninvasive models in predicting liver necroinflammation, the top three models for distinguishing G ≥ 2 in the HBeAg-positive group were Gao-1 (0.794), APGA (0.781), and Wang I (0.774). Similarly, for discriminating G ≥ 3, the best models were Wang I (0.894), APGA (0.878), and Gao-1 (0.873). For patients in the HBeAg-negative group, the most effective models for differentiating G ≥ 2 were APGA (0.785), Mehdi’s model (0.783), and the XIE model (0.77). Additionally, for discriminating G ≥ 3, the Xie model (0.897), AGAP (0.881), Logit(Y) (0.879), and APGA (0.873) demonstrated superior performance (Table 3).

### 3.4. Reassessment and Comparison of Noninvasive Models in Patients with Varying Levels of ALT below Two Times the ULN (ULN = 40 U/L)

Initially, we excluded patients whose ALT or AST levels exceeded 80 U/L. Subsequently, we assessed the diagnostic performances of the 55 noninvasive models in CHB patients with varying ALT levels below the ULN of 40 U/L. Among these models, the APGA, AGAP, GAPI, S-index, and XIE models exhibited significant differences in their ability to distinguish liver fibrosis and necroinflammation stages when ALT was below the ULN. Additionally, APGA, FIB-6, FI, and RPR demonstrated potential for a good performance when the ALT concentration exceeded the ULN (Appendix A).

### 3.5. Evaluation and Comparison of Noninvasive Models Developed in Cohorts with CHB versus Those in Other Chronic Liver Diseases (CLDs)

In the present article, we aimed to explore additional noninvasive models developed on CHB cohorts, including the straightforward models GP, RPR, and GPR [19], as well as some innovative, noninvasive models and nomograms such as the AA index, AGAP, AGPR, the IT model, PAPAS, APRG, APGA, APPCI, APPR, ATPI, eLIFT, Gao-model-1, Gao-model-2, GAPI, GqHBsR, HB-F, the HBeAg(+)model, INPR, Logit(Y) [33], mFIB-4, Mehdi’s model, PGA, S index, Wang I, Wang II, and Xie-model.

Finally, we evaluated 29 models by excluding those with parameters that could not be obtained from routine laboratory tests (Appendix A). Other than APGA, there were other models with excellent predictive value for SF and AF, with AUROCs exceeding 0.700. Among these, the four best-performing models were GAPI (0.719, 0.802), the XIE model (0.714, 0.8), AGAP (0.713, 0.794), and the S-index (0.708, 0.797). The performance of other models that were developed based on chronic hepatitis C, PBC or NAFLD cohorts varied in our study. However, GUCI, Fibro-α, FI, APRI, and King’s score performed better in our cohort, especially for AF patients, with AUROCs higher than 0.700. The AUROCs of the CHB cohort-based models, such as the APGA, GAPI, and XIE models, were greater than those of the recommended models, such as the APRI and FIB-4, for single patients, as shown in Figure 4.

### 3.6. Comprehensive Evaluation of Noninvasive Models

Given the variability of the noninvasive models with superior performances across different fibrosis stages, it is challenging to identify one or a few models that are unequivocally superior. To evaluate the noninvasive models, we adopted the scoring system developed by Dong [32] (Table 1). According to this grading system, only APGA was classified as grade A with high diagnostic value for discriminating both liver fibrosis stage and liver necroinflammation in all patients. Additionally, the other two noninvasive models, GAPI and the XIE model, also exhibited improved performance (Table 4 and Figure 4).

A Spearman correlation analysis (Table 5) was further employed to evaluate the associations between the serum marker levels, liver inflammation grade, and fibrosis stage. Both inflammation and liver fibrosis stages were found to be significantly correlated with the common parameters included in the noninvasive models or indices. Notably, among the three highly effective noninvasive models, the APGA index exhibited remarkably positive correlations with liver inflammation and fibrosis, with Spearman’s scores of 0.452 and 0.405, respectively.

The diagnostic performance of the top three models in our study for liver fibrosis and inflammation prediction is presented in Table 6, along with the corresponding cutoff values, sensitivity, specificity, NLR, NPV, PLR, and PPV. Additionally, the APGA index demonstrated a consistently favorable performance.

## 4. Discussion

The HBV infection remains a global public health challenge, and cirrhosis and HCC result in high morbidity and mortality rates. This issue has also had significant economic and societal impacts [34]. Recently, the indications for antiviral therapy have been expanded to include more chronic HBV patients, who require longer treatment courses before they meet the criteria for discontinuation. However, prolonged treatment can lead to concerns, such as asymptomatic patients starting medication without taking their illness seriously or adhering poorly to medication regimens, potentially leading to adverse drug reactions, drug resistance, high recurrence rates of hepatitis, serious liver failure requiring hospitalization or transplantation, or even death [35]. Therefore, it is important to assess the liver inflammation and fibrosis stage prior to initiating antiviral treatments to improve awareness of the disease and avoid these issues while benefiting patients.

In addition to liver biopsy data and recommended models, such as FIB-4 and the APRI index according to various guidelines, an increasing number of models based on CHB cohorts are being developed using both common clinical inspection indices and innovative indices, including MMP-1, PIIINP, sCD163, α2-MG, quantitative HBeAg and anti-HBc, as well as TE, 2D-SWE and the FibroTest. Models incorporating innovative indices consistently demonstrate a superior performance in distinguishing different stages of liver fibrosis. However, the implementation of new serum indices often requires additional detection methods that may be costly and operator dependent. Moreover, the markers utilized in these prediction models may not be readily available in routine non-research laboratories in China. Therefore, this study focused on evaluating models solely based on commonly used and easily obtainable serum clinical indices that have been previously employed for predicting liver fibrosis. Particular emphasis is placed on models derived from HBV cohorts.

In the present study, we included 55 models obtained through comprehensive searches of PubMed, CNKI, and other databases. The formulas or nomograms are presented in Appendix A. The majority of these models were developed based on routine laboratory parameters, including AST, AFP, ALP, ALB, GGT, PLT, INR and PTA; however, some models have incorporated quantitative measurements of HBsAg and HBV DNA.

Among the models used to predict liver fibrosis stages, a noninvasive predictive model called APGA, which was classified as grade A in our study (Table 4), was developed by James Fung [23]. The APGA model utilizes AST, platelet count, GGT, and AFP to predict both severe fibrosis and cirrhosis. In the original study, this model achieved an AUROC of 0.85. After conducting our own analysis on our cohort, we found that the model had high diagnostic value, with AUROCs of 0.750 and 0.832 for SF and AF, respectively. However, liver fibrosis and cirrhosis were measured using TE in previous studies, with cutoff values >8.1 kPa and >10.3 kPa, respectively. This may limit the correlation of the model results with actual liver histology. Seto [21] evaluated this model in another Chinese CHB patient cohort and obtained results consistent with ours regarding the accuracy of SF prediction. However, Erdogan et al. [36] evaluated the diagnostic value of this model in a Turkish cohort without achieving comparable results. When we divided patients into two groups based on HBeAg status and ALT levels, the APGA model demonstrated exceptional performance in assessing fibrosis levels across all populations as well as among patients with different HBeAg states or ALT levels. Furthermore, the model results were positively correlated with liver histology, as shown in Table 5. Additionally, compared to the other models assessed in this study, APGA exhibited better overall performance stability when predicting liver fibrosis across various populations while maintaining relatively higher sensitivity, specificity, and NPV, as presented in Table 6.

We further discussed and explored the discriminative value of noninvasive models for liver inflammation. Regardless of the HBeAg state and ALT level, APGA also exhibited strong discriminatory ability for G ≥ 2 and G ≥ 3, with AUROCs of 0.779 and 0.874, respectively. In addition to this model, another logistic regression model (LRM, referred to as the XIE model in this article) [37] was developed to predict liver necroinflammation in patients with hepatitis B e antigen negative CHB with normal or minimally elevated ALT levels. The XIE model achieved similar results, with AUROCs of 0.744 and 0.861 in our study (Table 3). The AUROC for predicting inflammation grade was greater in the HBeAg-negative group than in the HBeAg-positive group, primarily because this model was initially developed in HBeAg-negative patients. However, there is limited research available on the application of this noninvasive model for predicting liver fibrosis.

In this article, we conducted a comprehensive search for noninvasive models developed on CHB cohorts and ultimately evaluated 29 models (Appendix A) after excluding those that required parameters not obtainable through routine laboratory tests. Among all of the patients, we identified several models in addition to the APGA model mentioned above that exhibited excellent predictive value for liver fibrosis and inflammation, with AUROCs exceeding 0.700. Notably, GAPI and the XIE model (LRM) [37] were the two top-performing models according to our analysis (as shown in the Tables). Our findings were consistent with previous studies demonstrating that GAPI [38], a novel fibrosis index utilizing γ-glutamyl transpeptidase (GGT), age, platelet count, and international normalized ratio (INR), exhibited a superior performance in predicting SF and AF among CHB patients based on higher AUROC values obtained from our cohorts. In addition to these outstanding models, several other models have been developed specifically for CHB patients, such as the AA index [39], mFIB-4 score [40], IT model [20], and PNALT; however, their AUROCs were below 0.700, indicating a poorer predictive performance.

The other models based on CHC, PBC, or NAFLD patients exhibited disparate performances in the present study. Notably, the models Fibro-α [41], FIB-6 [42], Virahep-C [43], APRI [16], FCI [44], GUCI [45], and Forns [46] demonstrated a superior performance in our cohort, particularly for AF, with AUROCs exceeding 0.700. Conversely, the HGM-2 [47], FIB-5 [48], RLR [49], and NLR models exhibited poor performance, with AUROCs of less than 0.500. In the case of HGM-2, which incorporates PLT count, INR, ALP, and AST as predictors, the AUROC for predicting F ≥ 3 in the HCV/HIV infection group was reported to be 0.844 and 0.815 for the estimation and validation groups, respectively, in the original article, indicating a remarkable ability to predict AF [47]. However, this result is significantly greater than our findings among CHB patients, suggesting that this model developed based on CHC cohorts may not be applicable to CHB cohorts. Another study applied HGM-2 in HCV patients and obtained favorable results, with AUROCs of 0.809, 0.898 and 0.909 for SF, AF, and cirrhosis, respectively [50]. Similarly, AUROCs of 0.843 and 0.917 for AF and cirrhosis, respectively, were reported for another HCV cohort [51]. Prior to our study, no research had utilized this model to predict liver fibrosis stage in CHB patients, further supporting its potential unsuitability for such cohorts.

The APRI and FIB-4 scores have been highly regarded and extensively discussed in various major guidelines for the management of CHB. However, their diagnostic accuracies have not yielded consistent results across reported studies. Based on the data from this study, both APRI and FIB-4 demonstrated moderate diagnostic accuracy. When we attempted to utilize these models for distinguishing between different stages of liver inflammation, we observed that they exhibited superior discriminatory performance for liver inflammation compared to fibrosis, as evidenced by higher AUROC values (as shown in Table 3), which is consistent with previously reported findings in CHB patients [32,52,53]. In our study, although their performances were slightly inferior to other novel models for diagnosing fibrosis and inflammation (as presented in the tables) [31,54], when comparing the AUROC of the APGA model with these two models, APGA was found to outperform them in the evaluation of liver histology(Figure 4).

Despite the relatively large cohort, it is important to acknowledge the limitations of this study, including its single-center and retrospective nature. Inevitably, there may be bias resulting from missing data and selection bias, particularly admission bias. The differences between our findings and previously reported results could be attributed to variations in sample size, liver histology validation methods, upper limits of detection for quantifying HBsAg and HBV DNA, and differences in study subjects; all of these factors have the potential to impact the outcomes. Furthermore, we observed that the APGA model exhibited a superior and consistent ability to distinguish liver fibrosis and necroinflammation; however, further verification is still needed in other cohorts.

In conclusion, the results of the present study conducted in a large Chinese CHB cohort suggested that among the 55 noninvasive models for staging liver fibrosis, the APGA model demonstrated superior accuracy in staging both liver fibrosis and inflammation. Generally, models developed based on CHB cohorts outperform those developed on other chronic liver disease patients. The predictive value of these models may be influenced by HBeAg status and ALT levels; however, the APGA model consistently outperformed the other models in predicting liver fibrosis and inflammation. This model has the potential to reduce the reliance on liver biopsy for antiviral treatment guidance and significantly improve patient compliance. Although most models showed a satisfactory performance in our study, certain biases hinder their practical application. Therefore, further investigations are needed to develop innovative, noninvasive models with enhanced practicability for assessing liver staging and dynamic monitoring.

## Figures and Tables

**Figure 1 diagnostics-14-00456-f001:**
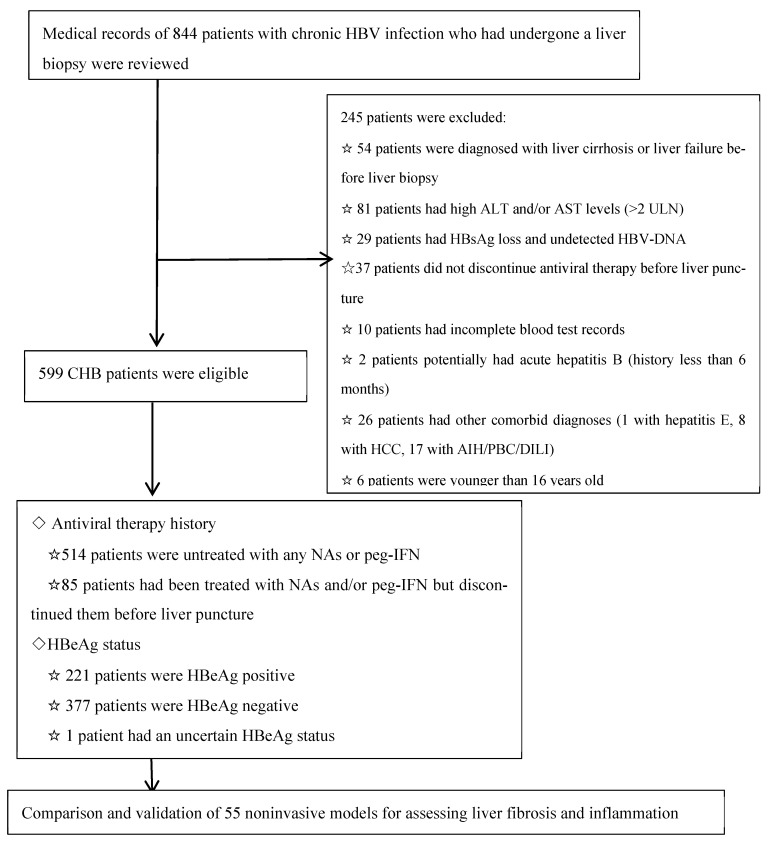
Flow chart of the study design. Abbreviations: AIH, autoimmune hepatitis; ALT, alanine transaminase; AST, aspartate transaminase; CHB, chronic hepatitis B; DILI, drug-induced liver injury; HBV, hepatitis B virus; HCC, hepatocellular carcinoma; NAs, nucleoside analogs; PBC, primary biliary cirrhosis.

**Figure 2 diagnostics-14-00456-f002:**
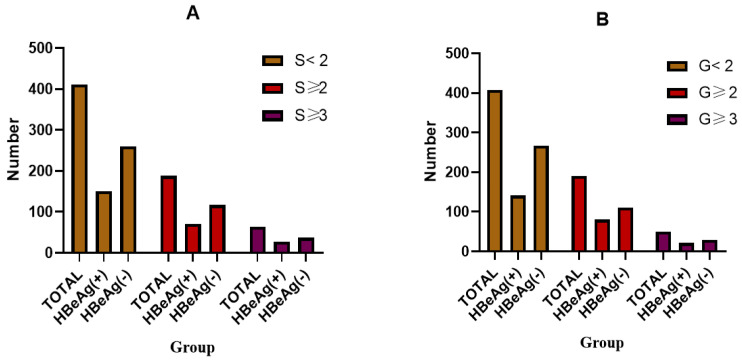
The incidences of liver fibrosis and inflammation in all patients and in the different groups ((**A**) liver fibrosis; (**B**) liver inflammation)).

**Figure 3 diagnostics-14-00456-f003:**
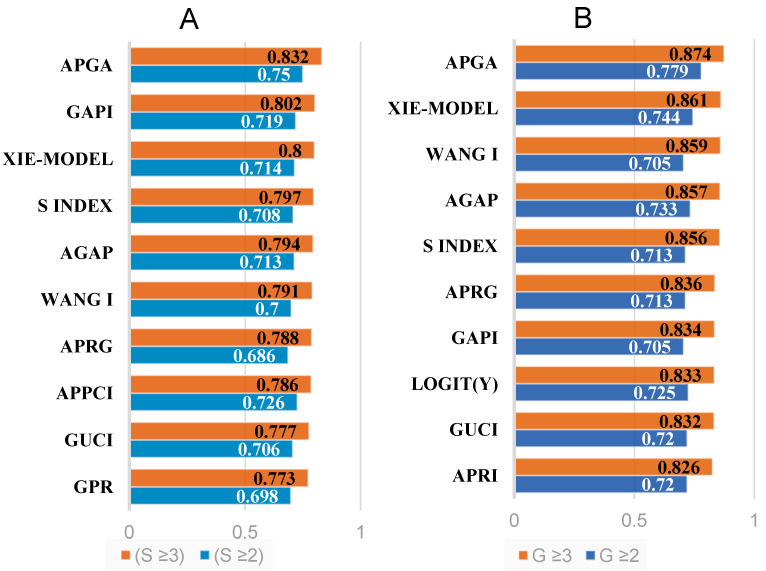
Of the 55 studied models, 10 models exhibited the best performance for evaluating liver fibrosis (**A**) and inflammation (**B**) in all patients.

**Figure 4 diagnostics-14-00456-f004:**
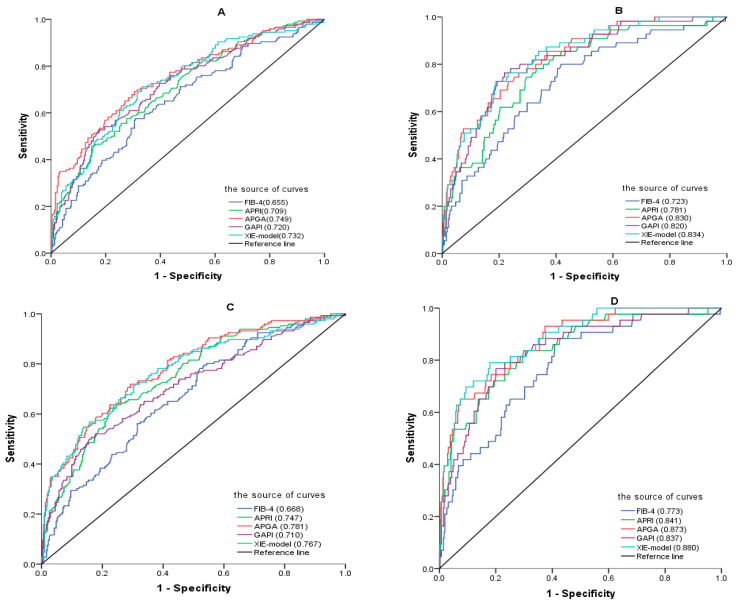
ROC curves of FIB-4, APRI, APGA, GAPI, and the XIE model for diagnosing liver fibrosis and inflammation in the same patient. S ≥ 2 (**A**), S ≥ 3 (**B**), G ≥ 2 (**C**), and G ≥ 3 (**D**).

**Table 1 diagnostics-14-00456-t001:** Grading system based on AUROCs derived from the data of this study for the evaluation of noninvasive models.

Histology Stages	Points			
	0	1	2	3
S ≥ 2 or G ≥ 2	<0.70	0.70–0.75	0.75–0.80	≥0.80
S ≥ 3 or G ≥ 3	<0.70	0.70–0.75	0.75–0.80	≥0.80

**Table 2 diagnostics-14-00456-t002:** Baseline characteristics of the enrolled patients.

Variables	Total (*n* = 599)	S0–1 (*n* = 411)	S2–4 (*n* = 188)	*p*-Value (S0–1 vs. S2–4)
Age, years #	37(31, 43)	36(31, 43)	37(32, 43)	0.357
Male, n (%)	411(68.6)	269(65.5)	142(75.5)	0.014 *
Log10[HBsAg], IU/mL #	3.36(2.88, 3.86)	3.32(2.86, 3.91)	3.43(3.0, 3.81)	0.656
HBeAg positive, n (%)	221(35.2%)	150(36.5%)	71(37.8%)	0.781
Log10[HBVDNA], IU/mL #	4.67(3.38, 6.81)	4.425(3.2975, 7.8025)	5.16(3.675, 6.11)	0.773
CP (mg/dL) #	0.201 ± 0.039	0.204 ± 0.037	0.197 ± 0.043	0.143
AFP (ng/mL) #	2.58(1.8, 4.0)	2.29(1.74, 3.50)	3.28(1.98, 5.98)	0.000 *
PT (s) #	13.3(12.9, 13.9)	13.2(12.8, 13.675)	13.55(13.1, 14.1)	0.000 *
PTA (%) &	97.8 ± 12.1	99.5 ± 12.4	94.0 ± 10.9	0.000 *
INR #	1.01(0.97, 1.06)	1.01(0.96, 1.05)	1.04(1.01, 1.09)	0.000 *
AST (U/L) #	26(21, 31)	24(21, 29)	28(24, 35)	0.000 *
ALT (U/L) #	31(22, 41)	28(21, 40)	36(25, 43.75)	0.001 *
GGT (U/L) #	22(16, 32)	20(15, 29)	28(19,38.75)	0.000 *
ALP (U/L) #	63(53, 76)	62(52, 73)	66.5(55, 83.25)	0.001 *
TBA (µmol/L) #	4.3(2.4, 8.6)	3.6(2.1, 7.7)	5.8(3.55, 12.55)	0.000 *
CHE (U/L) #	8616 ± 2047	8833 ± 2079	8135 ± 1893	0.000 *
Glu (mmol/L) #	5.04(4.64, 5.525)	5.04(4.65, 5.49)	5.02(4.6125, 5.49)	0.848
TCHO (mmol/L) #	4.81(4.3, 5.48)	4.85(4.32, 5.45)	4.77(4.22, 5.58)	0.743
ApoA1 (g/L) #	1.3(1.2, 1.5)	1.3(1.2, 1.5)	1.4(1.2, 1.5)	0.020 *
TBIL (µmol/L) #	10.3(7.5, 14.35)	10.1(7.2325, 14.675)	11(8.25, 13.8)	0.172
DBIL (µmol/L) #	2.8(1.9, 4.2)	2.6(1.8, 4.1)	3.2(2.3, 4.3)	0.001 *
ALB (g/L) &	46.1 ± 3.5	46.5 ± 3.2	45.3 ± 4.0	0.000 *
GLO (g/L) &	27.5 ± 3.8	27.2 ± 3.6	28.0 ± 4.0	0.01 *
PLT (×109/L) #	216(184, 250)	222(192, 258)	198(163, 231)	0.000 *
N (×109/L) #	3.25(2.59, 4.18)	3.36(2.64, 4.29)	3.115(2.4725, 3.9225)	0.035 *
L (×109/L) #	1.74(1.45, 2.12)	1.74(1.44, 2.09)	1.75(1.4725, 2.175)	0.364
RDW-SD (fl) #	40.2(38.4, 42.4)	40(38.3, 42.0)	40.75(38.525, 43.175)	0.015 *

# indicates the data are presented as the median (P25, P75); & indicates the data are presented as the mean ± SD; * indicates a significant difference. Abbreviations: HBsAg, hepatitis B surface antigen; HBeAg, hepatitis B e antigen; HBV DNA, hepatitis B virus DNA; CP, ceruloplasmin; AFP, alpha-fetoprotein; PT, prothrombin time; PTA, prothrombin activity; INR, international normalized ratio; ALT, alanine transaminase; AST, aspartate transaminase; γ-GGT or GGT, gamma-glutamyl transpeptidase; ALP, alkaline phosphatase; TBA, total bile acid; CHE, cholinesterase; Glu, glucose; TCHO, total cholesterol; ApoA1, apolipoprotein A1; TBIL, total bilirubin; DBIL, direct bilirubin; ALB, albumin; GLB, globulin; PLT, platelet; N, neutrophil count; L, lymphocyte count; RDW-SD, red blood cell distribution width-SD.

**Table 3 diagnostics-14-00456-t003:** Area under the ROC curve (AUROC) of the models for fibrosis stage and necroinflammation in the total population and different groups.

Noninvasive Models	Significant Fibrosis (S ≥ 2)			Advanced Fibrosis (S ≥ 3)			Significant Inflammation G ≥ 2			Advanced Inflammation G ≥ 3		
	Total (*n* = 188)	HBeAg (+) (*n* = 71)	HBeAg (−) (*n* = 117)	Total (*n* = 64)	HBeAg (+) (*n* = 27)	HBeAg (−) (*n* = 37)	Total (*n* = 191)	HBeAg (+) (*n* = 80)	HBeAg (−) (*n* = 110)	Total (*n* = 50)	HBeAg (+) (*n* = 21)	HBeAg (−) (*n* = 29)
AA index	0.600	0.559	0.621	0.684	0.681	0.684	0.630	0.573	0.662	0.673	0.662	0.677
AAR	0.533	0.581	0.505	0.594	0.689	0.529	0.488	0.517	0.479	0.597	0.697	0.529
AARPRI	0.616	0.644	0.599	0.687	0.752	0.643	0.577	0.592	0.581	0.710	0.753	0.682
ABA	0.570	0.607	0.549	0.651	0.696	0.636	0.596	0.634	0.594	0.689	0.692	0.709
AGAP	0.713	0.750	0.695	0.794	0.823	0.784	0.733	0.754	0.736	0.857	0.841	0.881
AGPR	0.698	0.710	0.691	0.768	0.812	0.743	0.701	0.724	0.696	0.812	0.801	0.827
ALRI	0.590	0.553	0.607	0.652	0.634	0.658	0.606	0.597	0.606	0.691	0.629	0.730
APGA	0.750	0.761	0.745	0.832	0.846	0.824	0.779	0.781	0.785	0.874	0.878	0.873
AP index	0.627	0.687	0.594	0.666	0.724	0.645	0.636	0.694	0.626	0.695	0.703	0.714
APRI	0.692	0.690	0.689	0.760	0.731	0.772	0.720	0.711	0.722	0.826	0.772	0.857
APPCI	0.726	0.682	0.757	0.786	0.806	0.777	0.720	0.717	0.725	0.737	0.737	0.736
APPR	0.670	0.668	0.674	0.721	0.773	0.692	0.673	0.690	0.670	0.761	0.741	0.784
APRG	0.686	0.701	0.678	0.788	0.812	0.776	0.713	0.729	0.714	0.836	0.811	0.863
ATPI model	0.649	0.628	0.658	0.737	0.740	0.732	0.650	0.655	0.646	0.771	0.742	0.788
CDS	0.634	0.649	0.625	0.707	0.749	0.678	0.613	0.615	0.617	0.716	0.759	0.687
Doha score	0.656	0.684	0.642	0.730	0.749	0.722	0.672	0.679	0.677	0.760	0.748	0.773
eLIFT	0.676	0.714	0.655	0.749	0.811	0.713	0.688	0.716	0.691	0.783	0.800	0.780
FCI	0.670	0.643	0.684	0.736	0.743	0.730	0.652	0.658	0.652	0.747	0.699	0.781
FI	0.662	0.677	0.650	0.761	0.758	0.763	0.694	0.670	0.716	0.821	0.795	0.839
FIB-4	0.659	0.717	0.630	0.726	0.790	0.698	0.672	0.719	0.671	0.769	0.797	0.769
mFIB-4	0.610	0.677	0.572	0.675	0.771	0.616	0.595	0.650	0.585	0.691	0.765	0.650
FIB-5	0.368	0.390	0.355	0.332	0.409	0.276	0.326	0.361	0.300	0.279	0.394	0.196
FIB-6	0.647	0.689	0.627	0.737	0.779	0.727	0.700	0.725	0.709	0.823	0.803	0.866
FibroQ	0.623	0.687	0.588	0.692	0.789	0.634	0.604	0.657	0.597	0.701	0.777	0.662
Forns	0.643	0.713	0.615	0.729	0.805	0.715	0.639	0.707	0.638	0.743	0.755	0.799
Fibro-α	0.674	0.696	0.665	0.746	0.818	0.705	0.649	0.670	0.654	0.773	0.854	0.720
GAPI	0.719	0.756	0.701	0.802	0.859	0.772	0.705	0.725	0.708	0.834	0.850	0.836
Gao-2	0.637	0.703	0.643	0.689	0.752	0.729	0.652	0.700	0.672	0.741	0.771	0.787
Gao-1	0.677	0.770	0.619	0.750	0.851	0.675	0.689	0.794	0.662	0.793	0.873	0.750
GqHBsR	0.650	0.695	0.633	0.684	0.774	0.643	0.632	0.653	0.634	0.728	0.784	0.707
GP	0.651	0.673	0.637	0.738	0.745	0.737	0.671	0.682	0.671	0.787	0.787	0.788
GPR	0.698	0.731	0.684	0.773	0.817	0.753	0.695	0.697	0.705	0.818	0.835	0.817
GUCI	0.706	0.700	0.706	0.777	0.750	0.787	0.720	0.712	0.719	0.832	0.782	0.857
HBeAg(+)model	0.637	0.703	0.619	0.689	0.752	0.675	0.652	0.700	0.662	0.741	0.771	0.750
HB-F	0.659	0.686	0.641	0.743	0.815	0.693	0.648	0.662	0.650	0.738	0.790	0.704
HGM-1	0.643	0.619	0.660	0.732	0.735	0.738	0.659	0.638	0.680	0.784	0.746	0.823
HGM-2	0.443	0.416	0.457	0.431	0.418	0.425	0.429	0.399	0.432	0.391	0.416	0.356
IT model	0.570	0.638	0.549	0.577	0.676	0.537	0.542	0.628	0.522	0.569	0.657	0.539
INPR	0.667	0.689	0.656	0.734	0.758	0.721	0.659	0.682	0.653	0.738	0.741	0.740
King’s score	0.690	0.718	0.677	0.754	0.767	0.754	0.719	0.747	0.714	0.807	0.773	0.839
Lok index	0.664	0.673	0.656	0.745	0.796	0.707	0.640	0.634	0.649	0.739	0.771	0.715
Logit(Y)	0.691	0.661	0.703	0.751	0.708	0.773	0.725	0.687	0.744	0.833	0.758	0.879
Mehdi’s model	0.592	0.370	0.718	0.619	0.350	0.751	0.646	0.417	0.783	0.680	0.376	0.858
NLR	0.453	0.427	0.469	0.496	0.499	0.499	0.426	0.455	0.408	0.449	0.399	0.488
NIKEI	0.566	0.636	0.526	0.629	0.740	0.567	0.557	0.634	0.539	0.648	0.751	0.596
PAPAS	0.639	0.697	0.616	0.670	0.731	0.653	0.675	0.746	0.674	0.700	0.731	0.704
PGA	0.542	0.553	0.536	0.560	0.623	0.513	0.532	0.539	0.528	0.606	0.600	0.613
PNALT	0.586	0.531	0.618	0.587	0.449	0.670	0.625	0.575	0.652	0.649	0.495	0.745
RPR	0.654	0.681	0.640	0.729	0.725	0.735	0.654	0.668	0.652	0.750	0.730	0.764
RLR	0.490	0.487	0.490	0.549	0.571	0.533	0.483	0.517	0.461	0.520	0.534	0.511
S index	0.708	0.737	0.695	0.797	0.827	0.782	0.713	0.702	0.732	0.856	0.864	0.858
Virahep-C model	0.652	0.687	0.636	0.707	0.768	0.686	0.694	0.737	0.693	0.771	0.737	0.826
Wang I	0.700	0.766	0.680	0.791	0.838	0.786	0.705	0.774	0.689	0.859	0.894	0.854
Wang II	0.695	0.725	0.678	0.758	0.809	0.726	0.663	0.678	0.658	0.769	0.769	0.772
XIE-model	0.714	0.670	0.739	0.800	0.769	0.816	0.744	0.696	0.770	0.861	0.796	0.897

Note: In this study, significant fibrosis (SF) and advanced fibrosis (AF) were defined as pathological stages ≥ S2 and ≥S3, respectively. Significant inflammation and severe inflammation were defined as pathological stages ≥ G2 and ≥G3, respectively.

**Table 4 diagnostics-14-00456-t004:** Grade distribution of the 55 noninvasive models for diagnosing liver fibrosis and necroinflammation.

Group	Grade A (9–12 Points)	Grade B (5–8 Points)	Grade C (0–4 Points)
Total	APGA	AGPR, APRI, APPCI, APRG, FI, FIB-6, GAPI, GPR, GUCI, King’s score, Logit(Y), S index, Wang I, XIE-model	AA index, AAR, AARPRI, ABA, ALRI, AP index, APPR, ATPI model, CDS, Doha score, eLIFT, FCI, FIB-4, mFIB-4, FIB-5, FIB-6, FibroQ, Forns, Fibro-α, Gao-2, Gao-1, GqHBsR, GP, HB-F, HBeAg(+)model, HGM-1, HGM-2, IT model, INPR, Lok index, Mehdi’s model, NIKEI, NLR, PAPAS, PGA, PNALT, RPR, RLR, Virahep-C model, Wang II
HBeAg(+)group	APGA, AGAP, GAPI, Gao-1Wang I	AGPR, APPCI, APRG, eLIFT, FIB-4 Forns, Gao-2, GPR, GUCI, S index King’s score, HBeAg(+) model, Wang II	AA index, AAR, AARPRI, ABA, ALRI, AP index, APRI, APPR, ATPI model, CDS, Doha score, FCI, FI, mFIB-4, FIB-5, FIB-6, FibroQ, Fibro-α, GqHBsR, GP, HB-F, HGM-1, HGM-2, IT model INPR, Lok index, Logit(Y), Mehdi’s model, NLR, NIKEI, PAPAS PGA, PNALT, RPR, RLR, Virahep-C model, XIE-model
HBeAg(−)group	APGA, XIE-model	AGAP, APRI, APPCI, APRG, FI FIB-6, GAPI, GPR, GUCI, Logit(Y), King’s score, Mehdi’s model, S index Wang I	AA index, AAR, AARPRI, ABA, AGPR, ALRI, AP index, APPR ATPI model, CDS, Doha score, eLIFT, FCI, FIB-4, mFIB-4, FIB-5 FibroQ, Forns, Fibro-α, Gao-2, Gao-1, GqHBsR, GP HBeAg(+)model, HB-F, HGM-1, HGM-2, IT model, INPR, Lok index, NLR, NIKEI, PAPAS, PGA, PNALT, RPR, RLR Virahep-C model, Wang II

Note: The final score of a model was the sum of the four scores. A model scoring 0–4 was designated Grade C, representing low diagnostic efficiency; a model scoring 5–8 was designated Grade B, representing moderate diagnostic efficiency; and a model scoring 9–12 was designated Grade A, representing high diagnostic efficiency.

**Table 5 diagnostics-14-00456-t005:** Correlations between noninvasive models or indices and the liver histology Spearman’s score.

Noninvasive Models or Indices	Inflammation Activity		Fibrosis Stage		Noninvasive Models or Indices	Inflammation Activity		Fibrosis Stage	
	Spearman’s r	*p* Value	Spearman’s r	*p* Value		Spearman’s r	*p* Value	Spearman’s r	*p* Value
AFP	0.291	<0.001	0.223	<0.001	GLB	0.0.95	0.021	0.116	0.005
PT	0.188	<0.001	0.224	<0.001	PLT	−0.245	<0.001	−0.233	<0.001
AST	0.304	<0.001	0.265	<0.001	N	−0.129	0.002	−0.086	0.035
ALT	0.207	<0.001	0.139	<0.001	RDW-SD	0.131	0.001	0.099	0.015
GGT	0.251	<0.001	0.255	<0.001	APGA	0.452	<0.001	0.405	<0.001
ALP	0.149	<0.001	0.153	<0.001	GAPI	0.331	<0.001	0.352	<0.001
DBIL	0.081	0.051	0.140	0.001	XIE-model	0.392	<0.001	0.344	<0.001
ALB	−0.224	<0.001	−0.125	<0.001					

Abbreviations: AFP, alpha-fetoprotein; PT, prothrombin time; ALT, alanine transaminase; AST, aspartate transaminase; GGT, gamma-glutamyl transpeptidase; ALP, alkaline phosphatase; DBIL, direct bilirubin; ALB, albumin; GLB, globulin; PLT, platelet; N, neutrophil count; RDW-SD, red blood cell distribution width-SD.

**Table 6 diagnostics-14-00456-t006:** Diagnostic performance of the top three models in predicting liver fibrosis and inflammation across all populations.

Models	AUROC (95% CI)	Cutoff	Se (%)	Sp (%)	PLR	NLR	PPV (%)	NPV (%)	*p* Value
S ≥ 2									
APGA	0.750(0.702–0.798)	6.72	69.1	69.8	2.2	0.4	52.2	82.5	<0.001
GAPI	0.719(0.674–0.763)	1.81	52.7	82.3	2.8	0.6	57.5	78.1	<0.001
XIE model	0.714(0.668–0.761)	−0.94	61.5	74.2	2.2	0.6	51.6	78.5	<0.001
S ≥ 3									
APGA	0.832(0.778–0.885)	7.27	76.8	73.4	3.1	0.3	28.4	95.8	<0.001
GAPI	0.802(0.746–0.858)	1.84	74.6	77.4	3.3	0.4	28.5	95.7	<0.001
XIE model	0.800(0.736–0.850)	−0.72	74.2	75.4	3.0	0.4	27.2	95.6	<0.001
G ≥ 2									
APGA	0.779(0.734–0.823)	6.72	70.8	70.8	2.3	0.4	52.2	83.6	<0.001
GAPI	0.705(0.658–0.751)	1.86	50.5	83.5	3.0	0.6	57.6	78.0	<0.001
XIE model	0.744(0.697–0.790)	−0.69	56.8	83.0	3.2	0.5	59.1	80.1	<0.001
G ≥ 3									
APGA	0.874(0.823–0.926)	8.53	65.9	91.8	7.1	0.4	42.1	96.3	<0.001
GAPI	0.834(0.776–0.892)	1.91	77.6	79.5	3.5	0.3	24.4	97.4	<0.001
XIE model	0.861(0.806–0.917)	−0.43	79.2	81.0	4.3	0.3	28.7	97.5	<0.001

Se, sensitivity; Sp, specificity; PPV, positive predictive value; NPV, negative predictive value; NLR, negative likelihood ratio; PLR, positive likelihood ratio.

## Data Availability

The data presented in this study are available on request from the corresponding author and the first author. The data are not publicly available due to the data will be uploaded to Research Data Deposit (RDD) of Sun Yat-sen University after publication.

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
