# Peer review of "Noninvasive Models to Assess Liver Inflammation and Fibrosis in Chronic HBV Infected Patients with Normal or Mildly Elevated Alanine Transaminase Levels: Which One Is Most Suitable?"

_diagnostics, 2024, doi:10.3390/diagnostics14050456_

Round 1
Reviewer 1 Report
Comments and Suggestions for Authors
This manuscript reported which noninvasive models were best suited to assess significant and advanced inflammation grade/fibrosis stage in chronically HBV-infected patients with normal or mildly elevated ALT levels. This study arouses interest for readers and provides an important clue to treat or observe such patients using a noninvasive grading system for liver necroinflammation and fibrosis. However, some issues should be addressed or modified.
(1) Most importantly, authors should specify and emphasize what is the ULN of ALT. The ALT ULN of 40 U/L should be described in the Methods (but not Results) section. In addition, authors should explain how this ULN value is determined with some references and how serum ALT levels were measured (e.g., assay kit). The ULN of 40 U/L appears slightly high; at least, this value is 30 U/L.
(2) How long did the patients have persistent normal or mildly elevated ALT levels? How often were the patients followed? In other words, authors should more clearly describe what definitions of “normal” and “mildly elevated” were (specifically, observation period but not one-time point)?
(3) Although subjects in this manuscript were patients with CHB, there is almost no information about HBV-related markers (e.g., HBsAg and HBV DNA levels). Why was no information provided about these crucial viral markers?
(4) Could you please specify the inclusion criteria in the Materials and Methods?
(5) As described in the text, 85 patients had received NAs and/or peg-IFN antiviral therapy but discontinued it for more than 1 year before liver biopsy. Why were these patients included? When such patients are excluded (i.e., treatment-naïve patients alone), will the results be the same or similar?
(6) As described in the text, Student's t-test and Mann-Whitney U test were employed for group comparisons. Which test was used to verify the distribution normality of variables?
(7) Table 1: Fully spelled terms for abbreviations had better be provided in the bottom note.
(8) There are too many typographic and grammatical errors. Authors should brush up on the manuscript more carefully.
Comments on the Quality of English Language
Please see the above (8).
Reviewer 2 Report
Comments and Suggestions for Authors
This manuscript showed that APGA could be the best non-invasive composited biomarker to evaluate fibrosis and inflammatory status in patients with chronic HBV infection who have no/mild elevation of ALT. Although the main methodology of the study is fine, the overall presentation of study result requires further improvement. Here are my suggestions.
1. The word “model” make the readers misunderstood on the objective of this study, since the model indicates about the representation or an example of something. I would suggest using marker or biomarker, or fully term as non-invasive, or serum, composited biomarker, instead.
2. AUROC is the only indicators to distinguish between “good” or “bad” biomarkers in the study. If the AUROC would be merely used, the author must try to proficiently explain to the ready why it is the best, how it is calculated, why do not use other indicators for the comparison?
3. Since there are overwhelming and similar abbreviations in the manuscript, it is quite difficult to follow the content. Therefore, all abbreviations would be nicely explained once it is initially used. An example is the “AUROC” in the abstract and NAs/CNKI in the material and methods. A summary part of abbreviation list might be needed.
4. Some texts in Figure 1 is missed.
5. What is the meaning of p-values in Table 1? Although it is the comparison between S0/1 to S2/4, this must be indicated.
6. What are the reasons to group patients in Table 1 to be S0/1 to S2/4? Why do not divide into S/G 0 1 2 3 4?
7. According to the previous suggestion (6), what is the difference in data presentation of Figure 2 when compared to Table 1?
8. Table 3b and 3c could be merged. Also, a better design would be needed since it’s difficult to differentiate A, B, and C.
9. Table 2, which showed after table 3???, should be improved. What is the meaning of bold, blue, red, underlined letters? Ideally, a single page with reduced letter size would be the choice.
10. Figure 3 A and B could be placed next to each other.
11. Since APGA are mentioned to be the best, also with other 2 top ranks and other references, a graph to show the correlation between biopsy results and APGA would be showed in a distribution pattern with the correlation coefficient.
12. If the previous suggestion (11) was done, Table 4 might be unnecessary. Also, most of the marker in Table 4 such as PT, ALT, AST, etc, are not one of 55 biomarkers. What are the reasons to compare them?
13. In general, the organization of tables and figure required extensive improvement.
14. If APGA is the best since the authors mentioned that “Which one is most suitable”, what is the pro- and con- of it.
